# The Neurochaperonopathies: Anomalies of the Chaperone System with Pathogenic Effects in Neurodegenerative and Neuromuscular Disorders

Federica Scalia [1,2], Alessandra Maria Vitale [1,2], Radha Santonocito [1], Everly Conway de Macario [3], Alberto J. L. Macario [2,3] and Francesco Cappello [1,2,*]

1   Department of Biomedicine, Neuroscience and Advanced Diagnostics (BIND), University of Palermo, 90127 Palermo, Italy; federica.scalia@unipa.it (F.S.); alessandramaria.vitale@unipa.it (A.M.V.); radha.santonocito@unipa.it (R.S.)
2   Euro-Mediterranean Institute of Science and Technology (IEMEST), 90139 Palermo, Italy; AJLMacario@som.umaryland.edu
3   Department of Microbiology and Immunology, School of Medicine, University of Maryland at Baltimore-Institute of Marine and Environmental Technology (IMET), Baltimore, MD 21202, USA; econwaydemacario@som.umaryland.edu
*   Correspondence: francesco.cappello@unipa.it or francapp@hotmail.com

**Abstract:** The chaperone (or chaperoning) system (CS) constitutes molecular chaperones, co-chaperones, and chaperone co-factors, interactors and receptors, and its canonical role is protein quality control. A malfunction of the CS may cause diseases, known as the chaperonopathies. These are caused by qualitatively and/or quantitatively abnormal molecular chaperones. Since the CS is ubiquitous, chaperonopathies are systemic, affecting various tissues and organs, playing an etiologic-pathogenic role in diverse conditions. In this review, we focus on chaperonopathies involved in the pathogenic mechanisms of diseases of the central and peripheral nervous systems: the neurochaperonopathies (NCPs). Genetic NCPs are linked to pathogenic variants of chaperone genes encoding, for example, the small Hsp, Hsp10, Hsp40, Hsp60, and CCT-BBS (chaperonin-containing TCP-1- Bardet–Biedl syndrome) chaperones. Instead, the acquired NCPs are associated with malfunctional chaperones, such as Hsp70, Hsp90, and VCP/p97 with aberrant post-translational modifications. Awareness of the chaperonopathies as the underlying primary or secondary causes of disease will improve diagnosis and patient management and open the possibility of investigating and developing chaperonotherapy, namely treatment with the abnormal chaperone as the main target. Positive chaperonotherapy would apply in chaperonopathies by defect, i.e., chaperone insufficiency, and consist of chaperone replacement or boosting, whereas negative chaperonotherapy would be pertinent when a chaperone actively participates in the initiation and progression of the disease and must be blocked and eliminated.

**Keywords:** chaperone system; molecular chaperones; chaperonopathies; nervous system; neurochaperonopathies; Hsps; neurodegeneration; neuromuscular disorders; chaperonotherapy



## 1. Introduction

A better understanding of neurodegenerative diseases may be achieved by examining them as part of the broad area of protein quality control, since they show signs of protein pathology. Indeed, a common feature in neuropathology is protein misfolding with subsequent formation of protein aggregates and precipitates, and various neurological disorders are proteinopathies or have a proteinopathy component [1–4]. In these, one or more proteins are pathogenic and prone to aggregate because of a genetic or acquired abnormality. In genetic proteinopathies, the gene encoding the pathologic protein bears a variant which makes its product pathogenic, whereas in the acquired proteinopathies the gene is normal, but the encoded protein may undergo a post-translation modification (PTM). In either case, aggregates formed by some of the abnormal proteins are cytotoxic

and cause disease. Through evolution, humans, as well as other organisms, have developed mechanisms to deal with proteinopathies. These mechanisms include protein-degradation and autophagic machineries, for instance the ubiquitin-proteasome system (UPS) [5–7] and chaperone-mediated autophagy (CMA) [8–10]. In addition, organisms have developed means to ensure that all proteins fold correctly, translocate to the place in which they function, resist the damaging effect of stressors, and regain activity after reversible damage and aggregation, namely all mechanisms that maintain protein homeostasis. Major players in protein-quality control are a variety of molecular chaperones, some of which are called Hsp (heat shock protein) that form groups of phylogenetically related proteins, such as the small-heat shock protein (sHsp), Hsp40(DnaJ), Hsp70(DnaK), Hsp90, and heavy Hsp families. In addition, there is a class of protein chaperones of about 60 kDa, also called chaperonins, of which two groups are distinguished, I (Hsp60, or Cpn60 for chaperonin 60) and II (CCT for chaperonin-containing TCP-1; also called TriC, for TCP-1 ring complex). All these chaperones typically function as multimolecular complexes of various sizes and degrees of complexity [11–17] and are part of the chaperone system (CS), which also includes co-chaperones, chaperone co-factors, and chaperone interactors and receptors [18–20]. The main functional partners of the CS in protein-quality control are the UPS and the CMA machineries. When the CS malfunctions, for example because of an abnormal chaperone, disease may ensue: a chaperonopathy [18,21] (free access updates at http://www.chaperones-pathology.org/). In many instances chaperonopathies have a pathogenic impact on the nervous system and contribute to the mechanisms underpinning neuropathies [20,22–24]. In this article, we will briefly discuss examples of neuropathies in which chaperonopathies play an etiologic-pathogenic role. These conditions we will call neurochaperonopathies (NCPs).

## 2. Types of Chaperonopathies in NCPs

NCPs are a heterogeneous group of diseases with complicated etiological and pathogenic features, among which failure of the CS can contribute to various degrees, depending on the disease and the patient considered. Here, we will briefly discuss some salient aspects of NCPs we considered useful to pathologists and physicians for understanding these conditions and for managing these patients.

The canonical functions of the CS pertain to maintenance of protein homoeostasis together with the CMA and the UPS, while its non-canonical functions are unrelated with the former and mostly co-involve the immune system (IS) [20,21]. A deficiency in a component of the CS, for example a chaperone, may cause disease, a chaperonopathy, which can be genetic or acquired [18,21]. In the former, a genetic variant, for example a mutation, results in the production of a defective, malfunctional chaperone, which may lead to tissue lesions and disease (illustrative examples and pertinent bibliography are provided in Table 1). Instead, in acquired chaperonopathies the gene encoding the affected chaperone is normal, but the gene product, namely the protein chaperone, is not. Chaperone abnormalities, genetic and acquired can also be classified considering the level of the chaperone's concentration and/or physiological activity by defect, excess, and mistake. In the chaperonopathies by defect, a chaperone deficit is the distinctive feature, which can be qualitative, quantitative or both. In the former, the affected chaperone may be at the right concentration, but its function is impaired because of a structural defect, for example; or the chaperone may be at a concentration below the normal range and it cannot reach the physiological level of functionality; or both deficits may coincide, below-normal concentration and functional ability, a situation often occurring in genetic chaperonopathies. In this regard, it should be emphasized that in neuropathies with abundant protein aggregates and precipitates, the level of chaperones available to deal with the excessive demand from misfolding clients may not be sufficient. This generates a chaperonopathy by defect caused by a disproportionate demand even when chaperone production is normal. The pool of useful chaperone molecules may be even further depleted by their being sequestered in the aggregates and precipitates [21] (free access

updates at http://www.chaperones-pathology.org/). The opposite parameters occur in the chaperonopathies by excess, namely the affected chaperone is either quantitatively above the normal range of concentrations or is qualitatively abnormal, displaying abnormal functions (gain of function).

The chaperonopathies by mistake include disorders in which a chaperone, apparently normal according to the results of tests currently available, plays a pathogenic role: it participates in the mechanism of disease as it occurs in various types of cancer and inflammatory and autoimmune conditions. In these instances, the pathogenic chaperone may have been structurally modified post-transcriptionally, for example by a post-translation modification (PTM), and thereby its intrinsic properties and functions changed. This is a field that calls for intensive research to develop means to detect PTMs and measure their impact on the chaperone molecule's properties and functions in single cells or using minimal amounts of chaperone. A case in point is the chaperone Hsp60, which plays a variety of roles, canonical and non-canonical, physiological, and pathogenic, even though there is only a single *Hsp60* gene in the human genome. How this multifunctionality is generated when there is only a single Hsp60 molecular type? One mechanism could be through post-translational modifications, a possibility that deserves further investigation [25].

**Table 1.** Examples of genetic neurochaperonopathies.

| Mol. Chap. | Mutation | Disease | M. I. | Ref. |
|---|---|---|---|---|
| **HSPA9** | Homozygous c.376C-T transition in exon 4 resulting in p.Arg126Trp substitution | EVEN-plus syndrome | AR | [26] |
| | Compound heterozygous for a c.383A-G transition in exon 4 resulting in p.Tyr128Cys substitution, and a 2-bp deletion in exon 8 (c.882_883delAG) causing a frameshift and resulting in a premature termination codon at amino acid 296 | EVEN-plus syndrome | AR | [26] |
| **HSPB8** | Heterozygous c.421A-G transition in exon 1 resulting in p.Lys141Glu substitution | dHMN2A | ADo | [27,28] |
| | Heterozygous c.423G-C transversion in exon 1 resulting in p.Lys141Asn substitution | dHMN2A | ADo | [28,29] |
| | c.423G-T transversion in exon 1 resulting in p.Lys141Asn substitution | CMT2L | ADo | [30,31] |
| **HSPB1** | c.100G-A transition in exon 1 resulting in p.Gly34Arg substitution | dHMN2B | Sporadic | [32] |
| | c.116C-T transition in exon 1 resulting in p.Pro39Leu substitution | dHMN2B/CMT2F | ADo | [32,33] |
| | c.121G-A transition in exon 1 resulting in p.Glu41Lys substitution | dHMN2B | ADo | [32] |
| | c.250G-C transversion in exon 1 resulting in p.Gly84Arg substitution | dHMN2B/CMT2F | ADo | [33–35] |
| | Homozygous c.295C-A transversion in exon 1 resulting in p.Leu99Met substitution | dHMN2B/CMT2F | AR | [33] |
| | c.379C-T transition in exon 1 resulting in p.Arg127Trp substitution | dHMN2B/CMT2F | ADo | [36,37] |
| | c.404C-G transversion in exon 1 resulting in p.Ser135Cys substitution | dHMN2B/CMT2F | ADo | [38,39] |
| | c.404C-T transition in exon 1 resulting in p.Ser135Phe substitution | dHMN2B/CMT2F | ADo | [33,36,40] |
| | c.406C-T transition in exon 2 resulting in p.Arg136Trp substitution | CMT2F | ADo | [36] |
| | c.407G-T transversion in exon 2 resulting in p.Arg136Leu substitution | dHMN2B/CMT2F | Sporadic | [32] |

**Table 1.** *Cont.*

| Mol. Chap. | Mutation | Disease | M. I. | Ref. |
|---|---|---|---|---|
| | c.418C-G transversion in exon 2 resulting in p.Arg140Gly substitution | dHMN2B/CMT2F | ADo or sporadic | [33] |
| | c.421A-C transversion in exon 2 resulting in p.Lys141Gln substitution | dHMN2B | ADo | [41] |
| | c.452C-T transition in exon 2 resulting in p.Thr151Ile substitution | dHMN2B | ADo | [36] |
| | c.490A-G transition in exon resulting in exon 3 resulting in p.Thr164Ala substitution | CMT2F | ADo | [42] |
| | c.539C-T transition in exon 3 resulting in p.Thr180Ile substitution | dHMN2B/CMT2F | ADo | [32,43] |
| | c.544C-T transition in exon 3 resulting in p.Pro182Ser substitution | dHMN2B | ADo | [44] |
| | c.545C-T transition in exon 3 resulting in p.Pro182Leu substitution | dHMN2B | ADo | [36] |
| | c.562C-T transition in exon 3 resulting in p.Arg188Trp substitution | CMT2F | Sporadic | [32] |
| **HSPE1** | Heterozygous c.217C-T transition in exon 2 resulting in p.Leu73Phe substitution | Undefined neurologic disorder | Sporadic | [45] |
| **DNAJB2** | Homozygous c.14A-G transition in exon 1 resulting in p.Tyr5Cys substitution | CMT2F/DSMA5 | AR | [46] |
| | Homozygous G-A transition in intron 4 (c.229+1G-A) | dHMN/DSMA5 | AR recessive | [46] |
| | Homozygous G-A transition in the donor splice site of exon 5 (c.352+1G-A) | DSMA5 | AR | [47] |
| **DNAJC5** | Heterozygous c.344T-G transversion in exon 3 resulting in p.Leu115Arg substitution | CLN4B | ADo | [48–51] |
| | Heterozygous 3-bp deletion in exon 3 (c.346_348del) resulting in p.Leu116del | CLN4B | ADo | [48–51] |
| **HSPD1** | Homozygous c.86A-G transition in exon 2 resulting in p.Asp29Gly substitution | HLD4 or MitCHAP-60 disease | AR | [52,53] |
| | Heterozygous c.292G-A transition in exon 3 resulting in p.Val98Ile substitution | SPG13 | ADo | [54] |
| | Heterozygous c.1381C-G transversion in exon 10 resulting in p.Gln461Glu substitution | SPG13 | ADo | [55] |
| **CCT5** | Homozygous c.440A-G transition in exon 4 resulting in a His147Arg substitution | Hereditary sensory neuropathy with spastic paraplegia | AR | [56,57] |
| | Homozigous c.670C>G transversion in exon 5 resulting in Leu224Val substitution | Demyelinating neuropathy with severe motor disability. | AR | [58] |
| **BBS6** | c.110A-G transition in exon 1 resulting in p.Tyr37Cys substitution | BBS | AR | [59,60] |
| | c.155G-A in exon 1 transition resulting in p.Gly52Asp substitution | BBS | AR | [61] |
| | c.169A-G transition in exon 1 resulting in Thr57Ala substitution | BBS | AR | [59] |
| | Homozygous 1-bp deletion (c.281del) in exon 2 resulting in a frameshift after amino acid Phe94 (p.Phe94fs), terminating the protein at amino acid 103 | BBS | AR | [59,61] |

**Table 1.** *Cont.*

| Mol. Chap. | Mutation | Disease | M. I. | Ref. |
|---|---|---|---|---|
| | Homozygosity for a complex 2-bp deletion (c.429_430del and c.433_434del) in exon 2 resulting in a frameshift and a premature termination of the protein at amino acid 157 | BBS | AR | [59,61] |
| | Nonsense mutation leading to premature termination (c.442C-T transition in exon 2 resulting in p.Gln148Ter) | BBS | AR | [60] |
| | c.792T-A transversion in exon 3 resulting in a premature termination (p.Tyr264Ter) | BBS | AR | [61] |
| | c.830C-T transition in exon 3 resulting in p.Leu277Pro substitution | BBS | AR | [59] |
| | c.1496G-C transversion in exon 3 resulting in p.Cys499Ser substitution | BBS | AR | [60] |
| **BBS10** | c.32T-G transversion in exon 1 resulting in p.Val11Gly substitution | BBS | AR | [62] |
| | c.101G-C transversion in exon 1 resulting in p.Arg34Pro substitution | BBS | AR | [63] |
| | 1-bp insertion in exon 2 (c.271dupT) leading to premature termination (p.Cys91fsTer95) | BBS | AR | [63,64] |
| | c.273C-G transversion in exon 2 resulting in p.Cys91Trp substitution | BBS | AR | [64] |
| | 4-bp deletion in exon 2 (c.909_912del) resulting in premature termination (p.S303fsTer305) | BBS | AR | [63] |
| | c.931T-G transversion in exon 2 resulting in p.Ser311Ala substitution | BBS | AR | [62,63] |
| | 2-bp deletion in exon 2 (c.1044_1045del) resulting in a frameshift and premature termination (p.Pro350fs) | BBS | AR | [64] |
| **BBS12** | Homozygous 3-bp deletion in exon 2 (c.337_339del) resulting in p.Val113del | BBS | AR | [65] |
| | Homozygous c.865G-C transversion in exon 3 resulting in p.Ala289Pro substitution | BBS | AR | [65] |
| | c.1063C-T transition in exon 3 resulting in a nonsense mutation (p.Arg355Ter) | BBS | AR | [65] |
| | 2-bp deletion in exon 3 (c.1115_1116del) resulting in frameshift and premature termination of the protein (p.Phe372fsTer373) | BBS | AR | [65] |
| | 2-bp deletion in exon 3 (c.1483_1484del) resulting in frameshift and premature termination (p.Glu495fsTer498) | BBS | AR | [65] |

Abbreviations: Mol. Chap., molecular chaperones; M.I., mode of inheritance; Ref., reference; ADo, autosomal dominant; AR, autosomal recessive; dHMN2A/2B, distal Hereditary Motor Neuropathy type 2A/2B; CMT2L/2F, Charcot-Marie-Tooth type 2L/2F; DSMA5, Distal Spinal Muscular Atrophy-5, CLN4B, Ceroid Lipofuscinosis Neuronal 4B; SPG13, Spastic Paraplegia 13; HLD4, Hypomyelinating leukodystrophy-4; BBS, Bardet–Biedl syndrome.

All the above-mentioned types of chaperonopathies, involving the chaperones listed in Table 1 have been implicated in neurological disorders and some examples of these NCPs are discussed below.

## 3. Genetic NCPs

### 3.1. Small Heat Shock Proteins

Diverse neurological and muscular disorders have been associated with mutations in sHsps. The sHsp α-crystallin family includes 10 members and one related protein (HSP16.2/HSPB11) [66] that, in addition to the chaperoning function, participate in cytoskeleton stabilization and possess anti-aggregation and anti-apoptotic activities [67–71].

Most of the mutations found in the sHsps are located in the highly conserved α-crystallin domain, which is an 80–100 amino-acid-long domain responsible for the association/dissociation of sHsp dimers and for the formation and stabilization of large multisubunit homo- and hetero-oligomers [72]. These mutations lead to protein aggregation and cell death. Two hypotheses have been proposed to explain the pathologic findings: the mutated sHsp molecules may acquire an intrinsic toxicity (gain of toxic function), or they may suffer a loss of function with abolition of their protein quality control activity [73]. Two representative examples are the mutations occurring in the *HSPB8* and *HSPB1* genes that have been associated with the neuromuscular disorders Charcot-Marie-Tooth (CMT) disease and distal hereditary motor neuropathies (dHMN) (Table 1). The term CMT is used to indicate a clinically and genetically heterogeneous group of hereditary motor and sensory neuropathies, characterized by degeneration of peripheral nerves and subdivided in two subgroups: demyelinating (CMT1) and axonal (CMT2) [74]. dHMNs are genetically heterogenous characterized by degeneration of distal lower motor neurons, resulting in muscle weakness and atrophy [75]. However, in many forms of dHMN there may also be minor sensory abnormalities and/or a significant upper-motor-neuron involvement. For this reason, there is often an overlap not only with axonal forms of CMT, i.e., CMT2, but also with juvenile forms of amyotrophic lateral sclerosis, Kennedy's disease, spinal muscular atrophy, and hereditary spastic paraplegia [76].

Various types of chaperonopathies can occur, as follows: CMT type 2L is caused by mutations in *HSPB8* and CMT type 2F is caused by mutations in *HSPB1*, while dHMN type IIA is caused by mutations in *HSPB8* and dHMN type IIB is caused by mutations in *HSPB1* [27–44]. Often, a single mutation in *HSPB1* or *HSPB8* gives rise to one of these conditions. However, in many cases the same mutation was found associated with both diseases, such as the p.(Lys141Asn) substitution in HSPB8 [28,31], and the p.(Ser135Phe) [38,54] and p.(Arg127Trp) substitutions in HSPB1 [36,37,77]. Moreover, there are also examples of mutations found in patients with an unclear phenotype, as shown in Table 1.

For both *HSPB8* and *HSPB1*, the identified mutations in most cases occur in the highly conserved α-crystallin domain, likely destabilizing the proteins structurally and functionally [78]. To elucidate the molecular mechanisms responsible for the association between mutated forms of HSPB8 and HSPB1 and neuromuscular disorders, in vitro studies have been performed. For instance, it was reported that the expression of HSPB8 mutants p.(Lys141Asn) and p.(Lys141Glu) in motor neurons resulted in a reduction in average length and number of neurites per cell, without inducing cell death. In contrast, these abnormalities were very moderate in sensory neurons, and absent in cortical neurons and glial cells [79]. Thus, motor neurons appear to be more sensitive to HSPB8 dysfunction compared to sensory neurons, as indicated by the predominant motor neuron phenotype in dHMN and CMT2L. Moreover, both HSPB8 mutant forms tend to associate with HSPB1 mutant proteins, forming intracellular aggregates [28].

Studies in vitro with HSPB1 mutants demonstrated that some of the substitutions affecting the α-crystallin domain reduced motor neuron viability, impaired neurofilament assembly, destabilized microtubules, and disrupted the axonal transport of specific cellular cargoes and of mitochondria [36,80–84], events that can be responsible for the premature axonal degeneration, typical of both CMT and dHMN [85]. In addition, in vivo studies have shown that overexpression of HSPB1 mutants in neurons is sufficient to cause pathological and electrophysiological changes typically observed in patients with motor neuropathy [86].

### 3.2. DnaJ(Hsp40)

Mutations in members of the Hsp40(DnaJ) family have been associated with neurologic and muscular disorders. Two examples are variants of the *DNJAB2* and *DNAJC5* genes, encoding members of the subfamilies B and C, respectively. *DNAJB2* is mainly expressed in neurons and plays anti-protein aggregation and neuroprotective roles, as shown in models of neurodegenerative disorders [87–89]. The mutation c.352+1G-A in

*DNAJB2* was found associated with autosomal recessive distal spinal muscular atrophy-5 (DSMA5), a type of dHMN with young adult onset and characterized by progressive distal muscle weakness and atrophy with gait impairment and loss of reflexes [47] (Table 1). This genetic variant reduced or abolished chaperone expression, with a consequent accumulation and aggregation of misfolded proteins, which would cause lower motor neurons degeneration [47]. The same *DNAJB2* variant was found in patients diagnosed with CMT2, and careful examination of phenotype and clinical evolution showed that pure motor impairment occurred early in the disease course, with no initial sensory symptoms, but sensory disturbances in the lower limbs appeared as the disease progressed [90]. Two other *DNAJB2* gene variants were found in two families (Table 1). One (c.229+1G-A) showed a dHMN phenotype, with signs of distal muscular atrophy and paresis; the other, causing the missense mutation p.(Tyr5Cys), showed a CMT2 phenotype, with a noticeable sensory involvement, extending the group of *DNAJB2*-related diseases to include sensory neuropathy [46]. However, another evaluation of the phenotypes associated with these two latter genetic variations has classified them as DSMA5 (for details see the corresponding page on ClinVar database).

DNAJC5, also known as cysteine string protein (CSP), is abundant in neuronal cells and is a key element of the synaptic molecular machinery [48]. Two different genetic variations, an amino acid deletion and an amino acid substitution, were found associated with autosomal dominant adult-onset neuronal ceroid lipofuscinosis-4B (CLN4B), also known as Kufs' disease [48–51]. These mutations affect two evolutionarily conserved leucine residues located in a conserved region of the cysteine-string domain that is involved in palmitoylation and membrane targeting of the protein [91]. CLN4B belongs to the group of neuronal ceroid lipofuscinosis (NCLs), a genetically heterogeneous group of at least nine neurodegenerative disorders, clinically characterized by progressive cognitive and motor impairment, visual impairment, epileptic seizures, and premature death. Despite the different ages of onset, all forms presented a common feature, i.e., the lysosomal accumulation of auto-fluorescent lipo-pigment in neuronal cells and peripheral tissues, which caused progressive and selective neurodegeneration and gliosis with secondary white matter lesions [92,93]. To clarify the molecular mechanisms responsible for the negative effects of the identified CSP mutations in neuronal cells, functional in vitro studies were performed. It was shown that both mutants had an abnormal intracellular localization and were less efficiently palmitoylated compared to the wild type protein. Moreover, they formed aggregates, causing CSP depletion that, in turn, could be responsible for the inhibition of synapse formation and synaptic transmission [49,94].

### 3.3. Hsp70(DnaK)

Pathogenic mutations in one of the members of the Hsp70(DnaK) family have been reported [26]. This chaperone family is composed of at least 17 members [95] but thus far only one of them, the mitochondrial mtHsp70, also named mortalin or HSPA9, has been found mutated and causing disease. Two mutations of this chaperone were associated with a syndrome very similar to CODAS (Cerebral, Ocular, Dental, Auricular, Skeletal Syndrome), which is caused by a mutation in the LONP1 gene, however it differs in that it also shows severe microtia, nasal hypoplasia, and other malformations. These characteristics are indicated by the name given to this new condition: EVEN-PLUS (from Epiphyseal, Vertebral, Ear, Nose, Plus associated findings) syndrome.

### 3.4. The Chaperonins

The best-known examples of genetic neurochaperonopathies are those involving members belonging to the family of chaperonins, including the group I chaperonin Hsp60 (HSPD1, of which a few pathogenic variants have been identified in humans) and the group II chaperonin CCT.

**Chaperonins of Group I.** The chaperonin HSPD1 is typically expressed within the mitochondrion where it assists the folding of proteins destined to the matrix together

with the co-chaperonin Hsp10 (HSPE1) [45,96,97]. HSPD1 plays a key role not only in the maintenance of protein homeostasis in mitochondria, but also in sustaining cellular viability, since complete loss of the protein impaired mammalian development and postnatal survival, as demonstrated by an in vivo model of mice homozygous for an inactivation of *HSPD1* gene [98]. In humans, missense mutations of the *HSPD1* gene have been found associated with severe and chronic nervous system diseases. These include: 1) a dominantly inherited form of Spastic Paraplegia (SPG13) affecting motor neurons with the longest axons in the spinal cord and caused by either one of two missense mutations occurring at different positions, i.e., p.(Val98Ile) and p.(Gln461Glu), [54,55,99] (Table 1); and 2) a recessively inherited hypo-myelinating leukodystrophy (HLD4), also known as MitCHAP60 disease, which is a fatal early-onset neurodegenerative disorder characterized by pronounced cerebral hypomyelination and is caused by a single specific missense mutation, i.e., p.(Asp29Gly) [52,53] (Table 1). In vitro and in vivo studies have shown that these mutations destabilized the chaperonin oligomeric complex and reduced its ATPase and folding activities compared to the wild type form [100,101]. In addition to these two well-known neurological conditions, many others of various degrees of severity may be caused by other *HSPD1* mutations that are known to occur in human genomes [97,102].

A point missense mutation, p.(Leu73Phe), in the gene encoding the co-chaperonin Hsp10 or HSPE1 (*hsp10* or *HSPE1* gene), has been found in a patient with a history of infantile spasms, hypotonia, developmental delay, a slightly enlarged liver, macrocephaly, and mild non-specific dysmorphic features [45] (Table 1).

The association between neurological disorders and *HSPD1/E1* genes mutations is related to the great sensitivity of neuronal cells to mitochondrial dysfunction, since they have a metabolism mainly based on oxidative phosphorylation [103]. In vitro studies have shown that the expression of HSPD1 variants alters mitochondrial morphology, dynamics, and functionality [104,105].

**Chaperonins of Group II.** This group includes the CCT family, which in humans is composed of nine subunits: 1 through 8, with two versions of subunit 6; plus, three chaperones named BBS6, BBS10, and BBS12, which are involved in cilia biogenesis; and two other evolutionarily related molecules, CCT8L1 and CCT8L2, of which little is known [106] (Figure 1).

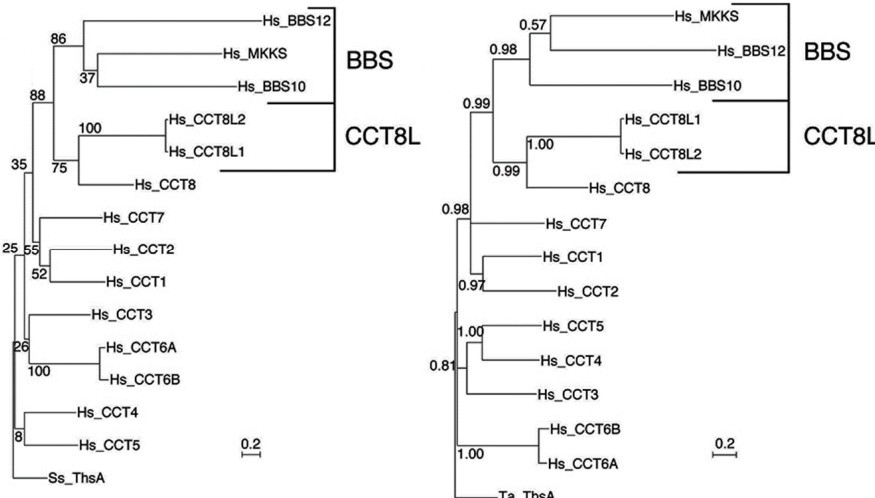

**Figure 1.** Evolutionary trees of chaperonin-containing TCP-1 (CCT) chaperonins. (**Left**): Maximum-likelihood tree of human chaperonins, including CCT monomers, MKKS, BBS10, and BBS12 as well as the members of the CCT8L class, CCT8L1 and CCT8L2. Numbers associated with each branch indicate bootstrap support from 100 replicates. Tree rooted by the archaeal thermosome α subunit of *Sulfolobus solfataricus* (Ss_ThsA). (**Right**): Bayesian tree of the same sequences. The numbers assigned to each branch indicate posterior probabilities. Tree rooted by the thermosome α subunit of *Thermoplasma acidophilum* (Ta_ThsA). Scale bars, number of substitutions per position for a unit branch length. Reproduced from Reference [106]

Here, we will consider chaperonopathies of the subunit CCT5 and those of the BBS cluster. BBS stands for Bardet–Biedl syndrome, which is the pathological condition caused by abnormalities in any one of these three chaperones. BBS6 is also called MKKS, because the disease associated with mutations in this gene is also called McKusick–Kaufmann syndrome.

The CCT subunits form functional hexadecamers with two octameric rings joined by their openings, which results in a barrel-like structure with a central cavity inside which polypeptide folding occurs [15]. This complex is also named TRiC, for TCP-1 ring complex, where TCP-1 stands for Tailless Complex Polypeptide 1. The integrity and functionality of this complicated multimolecular machine greatly depends on the integrity and functionality of all its individual components, the CCT subunits. For instance, a mutation in one of them may disrupt ring formation and hexadecamer assemblage or make the assemblage unstable and prone to dissociation, especially under the action of stressors.

A mutation occurring in subunit 5 of CCT was found to be associated with mutilating, distal sensory neuropathy [56,57] (Table 1). The functional CCT complex is composed of two hetero-octameric rings associated in a hexadecamer. The reported mutation, p.(His147Arg), occurred in subunit 5 (CCT5), and the disease was characterized by severe atrophy of the posterior tract of the spinal cord [56,57]. In vitro studies with human and archaeal purified molecules showed that the mutation, which occurs in the equatorial domain of the chaperonin subunit, impairs some of its properties and functions, resulting in poor oligomeric assembly [107–109].

More recently, a new disease was described associated with a different missense mutation, p.(Leu224Val), in CCT5 [58]. While the previously described mutation His147Arg caused a distal sensory neuropathy more pronounced in adulthood, the newly found mutation is associated with a motor neuropathy of early onset. Noteworthy is the different location of the mutations: His147Arg is in the equatorial domain, whereas Leu224Val is in the intermediate domain and appears to have an impact on the conformation of the apical domain, which is involved in substrate recognition and binding, namely functions quite distinct from those of the equatorial domain.

The disorders caused by mutations in the BBS genes affect ciliogenesis. Cilia are built with the participation of the BBSome, an octameric complex composed of seven BBS proteins—BBS1, BBS2, BBS4, BBS5, BBS7, BBS8, and BBS9—and the protein BBIP10 [110]. BBS6, BBS10, and BBS12 serve as chaperones for BBSome assembly. Defects in the process of ciliogenesis and, consequently, in cilia structure and/or function, lead to a group of disorders called ciliopathies [24,111]. The Bardet–Biedl syndrome is a genetic disorder with autosomal recessive inheritance, highly prevalent in inbred/consanguineous populations, and with numerous primary and secondary clinical manifestations, including obesity, retinal degeneration, olfactory deficits, nephropathy, polydactyly, development delay, and cognitive or other neurological impairments [24,112]. About 70–80% of all cases of Bardet–Biedl syndrome result from mutations in *BBS1* to *BBS18* genes, with frequent mutations in *BBS1* and *BBS10* genes, especially in populations of European and Caucasian descent [113].

The BBS proteins excluded from the BBSome, i.e., BBS6, BBS10, and BBS12, have been classified as chaperonins, since they have high sequence similarity with members of the CCT family [63,65,106,114] (Figure 1). Even if these proteins are not components of the BBSome, mutations in their encoding genes also lead to the same phenotype resulting from the loss or abnormalities of BBSome subunits [24,65,115]. Moreover, BBS patients bearing pathogenic variants in the *BBS6*, *BBS10,* and *BBS12* genes develop a more severe phenotype, with earlier disease onset and greater prevalence of all BBS primary diagnostic features typical of ciliopathies than patients with variants only in the non-chaperonin BBS genes [24,116]. The reason for this increased pathogenic severity associated with variants of the BBS chaperonin genes could be the malfunctioning of these three proteins, which are critical for BBSome assembly [24,117].

The Bardet–Biedl syndrome caused by mutations in the genes encoding for the chaperonin-like BBS proteins can be included in the group of genetic chaperonopathies [24],

and in particular within the neurochaperonopathies if we consider the associated neurological abnormalities. In a mouse model of BBS, the morphological evaluation of brain neuroanatomy revealed ventriculomegaly of the lateral and third ventricles, thinning of the cerebral cortex, and reduced volume of the corpus striatum and hippocampus [118]. These abnormalities could be related to defects in the cilia of the ependymal cells, affecting cilia assembly, structure, and/or function [118,119].

Numerous genetic mutations in the genes encoding BBS6, BBS10, and BBS12 have been found [24,59–65,120]. Illustrative examples are listed in Table 1. Mutations in the *BBS6* gene have been associated with typical BBS and with another similar disorder, the McKusick–Kaufman syndrome, and it was suggested that both syndromes are different allelic forms of the same clinical entity [59,121,122]. Moreover, in different families affected by BBS, many of the genetic variants in the *BBS6* gene were found in the same individual [61] or associated with mutations in genes encoding BBS2 and other BBS proteins, suggesting a triallelic inheritance model for penetrance of the BBS phenotype [60,120]. *BBS10* is, together with *BBS1*, the major contributor to BBS [113]. The most common mutation in the *BBS10* is the 271dupT, occurring in 46% of mutant alleles [63].

Many of the *BBS10* genetic variants were found in the same individual [62,64] or associated with mutations in the *BBS1* gene [63]. Variants of *BBS12* account for 8%–11% of the total cases of BBS in most of the cohorts reported [115].

## 4. Acquired Neurochaperonopathies

### 4.1. Hsp70(DnaK)

The Hsp70 family is composed of at least 17 members [95], and they are targets of a variety of PTMs such as phosphorylation, acetylation, ubiquitination, AMPylation, and ADP-ribosylation [123]. The only one human Hsp70 AMPylator enzyme containing Fic (Filamentation induced by cAMP) domain is the FICD enzyme, also known as Huntingtin yeast partner E (HYPE), able to catalyze the transfer of AMP onto a serine, threonine, or tyrosine residue of the substrate protein [124]. The consequences of Hsp70 AMPylation are poorly understood and, although there is no information on its impact on neurodegeneration to date, some interest in this regard has developed because of observations made using *Saccharomyces cerevisiae*, a eukaryote free of endogenous AMPylation machinery, and *Caenorhabditis elegans* models manipulated for the FIC-1 enzyme orthologous of human FICD (or HYPE) [125]. Induction of *C. elegans* FIC-1 (E274G) and *Homo sapiens* HYPE (E234G) in *S. cerevisiae*, in absence of stress, inhibited Hsp70 activity leading to decreased cell growth, toxic protein aggregation, and misfolding. These results suggest that AMPylation somehow modulates the Hsp70 protein folding machinery. Furthermore, other experiments confirmed AMPylation of a human Hsp70 family member, HSPA5, in a FICD-dependent manner, indicating that a high number of neuronal proteins involved in neuronal differentiation are AMPylated by FICD enzyme [126]. In addition, the fly *Drosophila*'s dFic AMPylator was shown to be essential for maintaining the levels of AMPylated Hsp70 BiP (the Hsp70 family member that resides in the endoplasmic reticulum (ER)), which are crucial for the photoreceptor's resistance against stress; flies deprived of dFic are blind and unable of postsynaptic activation [123].

On the other hand, *C. elegans* models for Alzheimer's disease (AD) and Parkinson 's disease (PD) showed the formation of large protein aggregates when AMPylation of three Hsp70 family members was FIC-1 induced [127] (Table 2). Strikingly, the larger aggregates were more strongly cytoprotective than the smaller ones, extending the nematode's life.

**Table 2.** Effects of PTM on molecular chaperones produced by in vivo and in vitro wild type and disease models.

| Model | Status | Molecular Chaperone | Human Ortholog | PTM | Effect | Ref. |
|---|---|---|---|---|---|---|
| *Drosophila* | Wt | Bip | Grp78 (HSPA5) | AMPylation | Blindness | [123] |
| *Saccha-romyces cerevisiae* | Wt | Hsp70 | HSPA | AMPylation | Decreased cell growth; protein misfolding; toxic protein aggregation | [125] |
| *Caeno-rhabditis elegans* | AD; PD | HSP-1; HSP-3; HSP-4 | HSPA | AMPylation | Large cytoprotective protein aggregates | [127] |
| PC12 cells | Wt | Hsp90 | HSPC | Nitration | ALS | [128,129] |
| PC12 cells | Wt | VCP | p97 | Phosphorylation and acetylation | Neurite retraction and shrinkage | [130] |

Abbreviations: PTM, post-translation modification; Ref., reference; Wt, wild type; Hsp or HSP, heat shock protein; AD, Alzheimer's disease; PD, Parkinson's disease; ALS, amyotrophic lateral sclerosis; VCP, valosin-containing protein.

### 4.2. Hsp90 (HSPC)

The human Hsp90 (HSPC) family is composed of various members: two are cytosolic, one resides in the ER, and the other in the mitochondrion (references in [21]). Hsp90 chaperones have multiple sites for phosphorylation, acetylation, SUMOylation, methylation, O-GlcNAcylation, ubiquitination, and other PTMs [128]. Nitration of tyrosine has been associated with apoptosis of motor neurons in amyotrophic lateral sclerosis (ALS) patients and superoxide dismutase (SOD)-transgenic animal models [129] (Table 2). Nitration of five of the 24 tyrosine residues of HSPC2 and HSPC3 impaired ATPase chaperone activity. Nitrated tyrosine at position 56 of HSPC3 was identified in spinal cord sections of an ALS patient and of a mutated mouse model. The nitrated HSPC induced cell apoptosis by activation of $P2 \times 7$ receptor-mediated calcium influx which induced Fas pathway in PC12 cells [129]. Several chronic and acute conditions causing oxidative stress induced the nitration of Hsp90 family members. However, the impact of Hsp90 nitration depends on the position of the nitrated tyrosine. In PC12 cells and motor neurons from rat embryo, it was observed that nitration of tyrosine at position 56 causes a gain of function, which is cytotoxic, whereas nitration of tyrosine at position 33 impairs mitochondrial activity [131].

Hsp90 collaborates with the Cell Division Cycle 37 (Cdc37) co-chaperone, forming a HSPC/Cdc37 complex, which binds almost 60% of the kinome preventing aggregation of activated regulators [132]. The co-chaperone Cdc37 is also a target of PTMs [133]. Phosphorylation at serine 13 of Cdc37 by CK2 (casein kinase II) affects the Hsp90/Cdc37 complex in performing its kinases homeostasis activity. In addition, the casein kinases 1 and 2 which involved AD, PD, and ALS pathogenesis have been reported to impair the interaction between Hsp90 and Cdc37 [134,135].

### 4.3. VCP/p97

The valosin-containing protein (p97 or VCP) is a homohexameric AAA+ ATPase protein of about 97 kDA. It is an abundant and ubiquitous molecule involved in cell cycle regulation [136], autophagy [137], DNA repair [138], and ER-associated and ubiquitin-mediated protein degradation [139]. VCP has been found within protein aggregates in PD, ALS, AD, and in polyglutamine aggregates [130]. Polyglutamine disease is characterized by deacetylation of histones H3 and H4 and a correlation of this deacetylation with VCP activity has been suggested [130]. VCP was found in the PC12 cell nucleus in protein aggregates with polyglutamine expansions, showing PTM, such as phosphorylation at Ser-612 and Thr-613, and acetylation at Lys-614 [130]. Nuclear, post-translationally modified VCP is associated with neurite retraction and shrinkage, features which have been found also in PC12 cells expressing protein aggregates with polyglutamine expansions [130]. Therefore, it has been suggested that VCP could be a mediator of histones H3 and H4 deacetylation, even if the underlined mechanism remains to be elucidated. VCP appears to be essential for the clearance of "occasionally" aggregated proteins, promoting the

deacetylation of H3 and H4 and, in turn, reducing the transcription of de novo proteins; however, during chronic and prolonged protein aggregation, nuclear translocation and post-translational modification of VCP could induce neurodegeneration and cause an atrophic phenotype as observed in the *Drosophila* experimental model [130].

## 5. Discussion

In the last few years, an increasing number of scientific reports have highlighted the involvement of members of the CS in the pathogenesis of diverse neurological disorders. It was found that genetic and/or acquired malfunction (or lack of function) of members of the CS are implicated in the pathogenic mechanisms underlying NCPs (Figure 2).

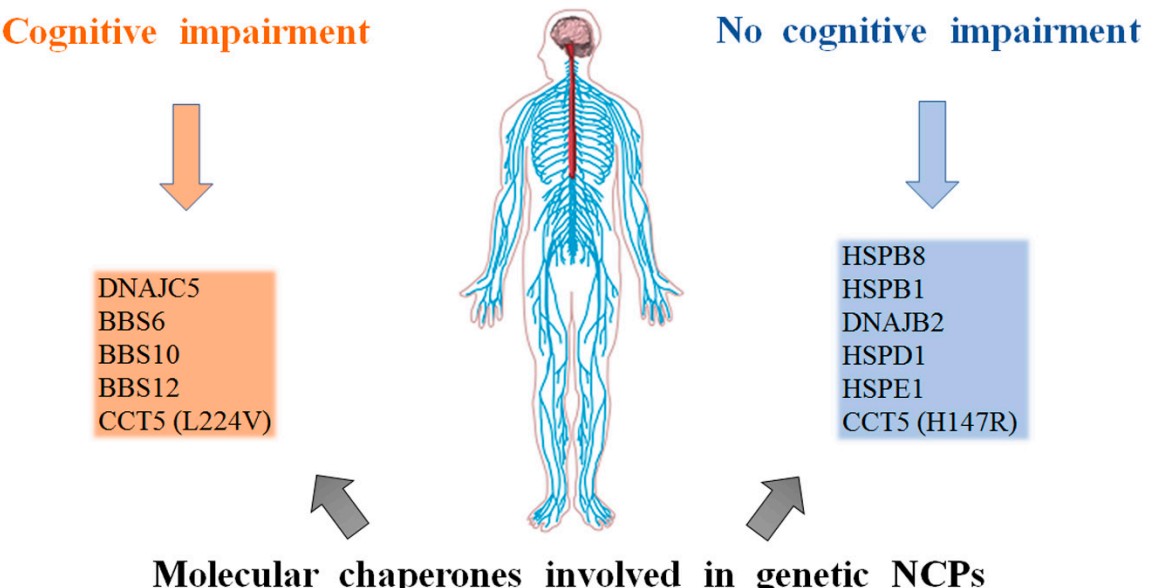

**Figure 2.** Molecular chaperones involved in genetic neurochaperonopathies (NCPs). Various molecular chaperones are involved in genetic NCPs, but only some (orange) are associated with cognitive impairment, while others (blue) are not.

The heterogeneity of these disorders and the difficulties implied in the study of the nervous system in humans are the major challenges for physicians and other investigators willing to elucidate the relevant molecular mechanisms and to develop efficacious treatment strategies. The data reported thus far, some of which has been briefly discussed in the previous sections, indicate that pathogenic variants of members of the sHsp, DnaJ(Hsp40), and CCT families, and of the Hsp60 gene, tend to cause predominantly motor disorders, although sensory and mental functions may also be affected [24,28,32,33,36,46,49,52,54]. For example, a missense mutation of the subunit number 5 of CCT, CCT5, was reported to be accompanied by a phenotype characterized mostly by distal sensory mutilating neuropathy [56,57]. In contrast, another recently discovered mutant located in a different structural domain of CCT5 was associated with a motor distal neuropathy without mutilation [58]. Thus, the impact of the mutation on the properties and functions of the chaperone molecule, and the accompanying tissue abnormalities and lesions observed in patients may greatly differ, depending on the location of the mutant amino acid, but other elements, e.g., environmental and nutritional factors that may also contribute to the generation of diverse phenotypes cannot be excluded.

Other genetic NCPs discussed are caused by variants of a subfamily of CCT genes composed of the BBS6 (MKKS), BBS10, and BBS12 genes [24,59–61,63–65]. Cognitive deficit was recorded only in patients with pathogenic variants of the CCT5 (the newly reported mutation p.(Leu224Val)), *DNAJC*, and *BBS* genes (Figure 2).

Only one of the members of the Hsp70(DnaK) family, the mitochondrial Hsp70 (mtHsp70 also named mortalin or HSPA9) has been reported with pathogenic mutations causing a syndrome that can be classified as NCP. Other members of this family, such

as the HSPA1A and HSPA1B, whose genes are organized in tandem within the so called *HSPA1*-cluster [140] are stress-inducible and have been implicated in the pathogenesis of several diseases, such as cancer and neurodegenerative conditions, but their genes were normal [141,142]. This pathogenicity of the Hsp70 molecules fits the description of acquired chaperonopathies by mistake or collaborationism, in which apparently normal chaperones help the disease rather than protect from it. The lack of genetic NCPs associated with mutations in *HSPA1A* and *HSPA1B* genes could be related to their vital roles suggested also by their evolutionary conservation [95]. This is also suggested by the fact that the majority of polymorphisms identified in the human *HSPA1*-cluster are in the UTR regions, and the most common SNPs occurring in the coding regions are synonymous and thus, do not affect the proteins' amino acid sequences [143,144]. A few studies experimentally investigated the impact of human *HSPA1A-1B* mutations on the function/expression of the encoded proteins [145–148]. The functional analysis of six naturally occurring variants of the *HSPA1A* gene revealed that these variants result in significant functional alterations, affecting ATP-hydrolysis rate and substrate-binding ability, likely by altering the allosteric communication between its two major domains. However, they are rare mutations, with very low penetrance in the human population. All these results support the notion that the *HSPA1A* gene is under strong selection and that its functions must be conserved [149], which would explain why no patients with pathogenic mutations of this gene have been found.

The Hsp90 (HSPC) family, including five members and various isoforms expressed in specific cellular organelles, is highly conserved from early eukaryotes to humans and plays important roles in protein homeostasis (references in [21]). Hsp90 chaperones have a wide array of client proteins and, together with their co-chaperones and co-factors, participate in cell survival, cell-cycle control mechanisms, hormone signaling and other signaling pathways, and apoptosis [150]. In addition, Hsp90 has been implicated in the pathogenesis of Alzheimer's disease and Parkinson's disease [151,152]. However, it is noteworthy that no Hsp90 mutations have been described yet as an etiological-pathogenic factor in NCPs. The same message is given by analysis of Hsp90 genetic polymorphisms. Polymorphic variants in introns and exons of the human Hsp90beta (*HSPC3)* gene have been detected with no apparent effects on phenotype [153].

In recent years, along with genetic NCPs, attention has also been directed to acquired NCPs associated with molecular chaperones with PTMs. For instance, in vivo studies demonstrated that changes in the AMPylation balance of molecular chaperones could influence cell fate in physiological [125] and pathological conditions, such as PD and AD [127]. Human FICD AMPylates various amino acids in the Hsp70, DNAJB1, and Hsp90 chaperones. AMPylation levels regulate the correct neurogenesis and neuronal conduction [123,126]. So far there are no data suggesting that chaperone AMPylation may be disruptive and contribute to the pathogenic mechanism of neurological disorders.

Nitration of Hsp90 (HSPC) family members in relation to ALS disease is an example of the differential outcomes one can expect depending on the site and level of the PTM. Furthermore, considering the importance of mitochondrial activity in neuronal cells, one may assume that nitrated Hsp90 may be associated with acquired NCPs in addition to those for ALS. Acquired NCPs could be the result of PTMs in molecular chaperones and of PTMs in co-chaperones, chaperone co-factors or chaperone interactors, as illustrated by the consequences of phosphorylation of Hsp90/Cdc37 complex [133,135].

Two coexisting PTMs, phosphorylation and acetylation, in the VCP in the nucleus may promote histone deacetylation [138]. VCP could also undergo methylation and S-glutathionylation, two modifications that occur during oxidative stress with impairment of ATPase activity [139]. Since oxidative stress is a prevalent feature of neurodegenerative diseases, it is possible that modified VCP also plays a pathogenic role in NCPs.

Pathogenic chaperones with aberrant PTMs associated with NCPs, other than those described above, have not been reported. Most of the naturally occurring or induced PTMs in Hsp60 (HSPD1) with a pathogenic role have been implicated in carcinogenesis [154–157],

apoptosis signaling regulation [158,159], and immune system regulation [160], but none has been associated with NCPs [25,161]. However, an involvement of Hsp60 (HSPD1) in AD and PD is under scrutiny. It is possible that Hsp60 with aberrant PTMs and/or with abnormal levels and location, i.e., intra- or extracellular, and when intracellular intra- or extramitochondrial, may be pathogenic in neurological disorders [25].

The number of multiple PTMs that can occur on a molecular chaperone is strikingly high, e.g., 60% and 56% of the T, Y, and S residues of Hsp70 (HSPA8) and Hsp90, respectively, could be phosphorylated [123] (Figure 3).

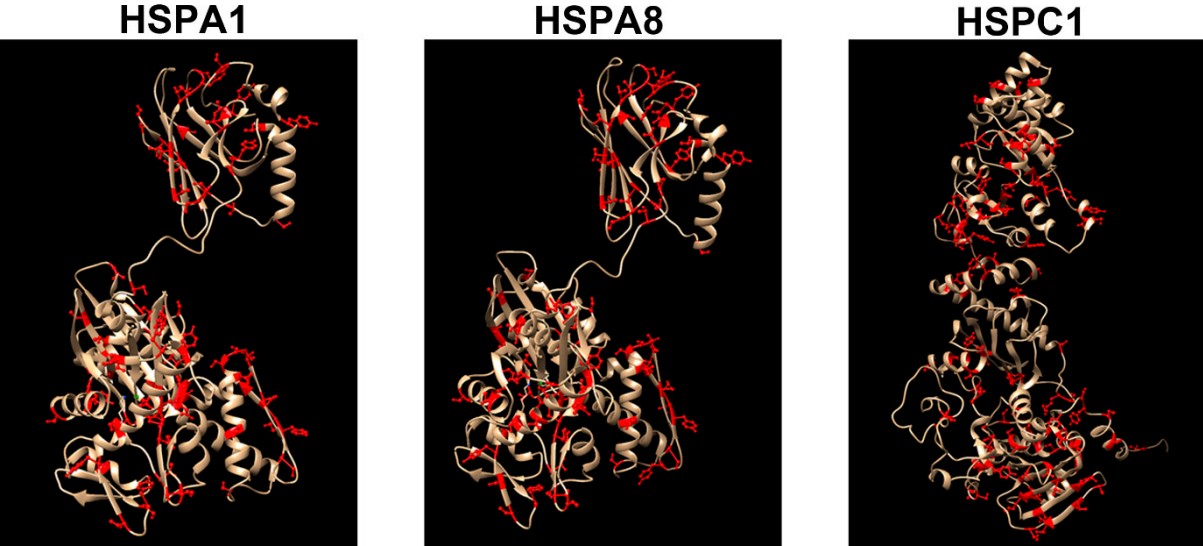

**Figure 3.** Examples of Hsp70(DnaK) and Hsp90 (HSPC) molecular chaperones rendered as light-brown ribbon models, showing the amino acids serine, threonine, and tyrosine in red (ball-and-stick). The models were built using SWISS-MODEL, developed by the Computational Structural Biology Group at the SIB Swiss Institute of Bioinformatics at the Biozentrum, University of Basel (https://swissmodel.expasy.org/, [162]), and were visualized with UCSF Chimera, developed by the Resource for Biocomputing, Visualization, and Informatics at the University of California, San Francisco, with support from NIH P41-GM103311 (https://www.cgl.ucsf.edu/chimera/, [163]). The HSPA1 and HSPA8 models were built using the NP_005336.3 and NP_006588.1, respectively, amino acid sequences reported in the NCBI Protein database, and as template the B chain of the crystal structure PDB ID 3C7N in the RCSB Protein Databank. The structures shown cover the HSPA1 and the HSPA8 sequences between amino acids 3 and 547. The HSPC1 model was built using the amino acid sequence NP_001017963.2 reported in the NCBI Protein database, and as template the B chain of the crystal structure PDB ID 5FWM in the RCSB Protein Databank. The structure shown covers the HSPC1 sequence between amino acids 137 and 821.

In addition, the same residue could be subjected to different PTMs, and several PTMs can also occur on the relevant co-chaperones and collaborators. All these modifications create the "Chaperone Code" that regulates chaperone functions, physiologically and pathologically (e.g., in cancer, viral infections, chronic inflammation, and neurodegenerative disorders). This flexibility and diversity of potential functions in health and disease of a single chaperone type, together with the variety of chaperone types, opens exciting novel avenues for investigating the mechanisms involved in acquired NCPs, for example those characteristics of ageing and AD and PD [135].

## 6. Conclusions and Future Perspective

Why is it worthwhile to determine if any given neurological patient bears a chaperonopathy? Firstly, the CS is present in all cells, tissues, and organs, therefore, a deficiency in any of its components may have a pathogenic impact in various anatomic areas and physiological systems, including the nervous system, and the probability of causing signs and symptoms is high. Secondly, the CS participates in a variety of physiological mechanisms, both as guardian of protein homeostasis and protector against stressors, and as

the effector of many other cellular processes unrelated to protein quality control, i.e., non-canonical functions, including critical interactions with the immune system. This also increases the probability of deficiencies in the CS having clinical manifestations. Thirdly, detection of a chaperonopathy opens the door to investigate the possibility of applying chaperonotherapy in any of its modes, which is a significant prospect considering that most chaperonopathies are serious diseases still awaiting efficacious treatment [20,164–167]. It would be beneficial for many patients if physicians would bear in mind the concept of hidden chaperonopathies, particularly in those cases in which the clinical signs and symptoms do not quite fit within the expected clinical picture for any given neurological disorder [168]. To unveil this hidden pathogenic factor, physicians should add a search for a chaperonopathy to the differential diagnosis algorithm.

**Author Contributions:** Project planning, F.C. and F.S.; literature searches, A.M.V., R.S., F.S., E.C.d.M. and A.J.L.M.; molecular modeling, F.S. and A.M.V.; writing, F.S., A.M.V., R.S., E.C.d.M. and A.J.L.M. All authors have read and agreed to the published version of the manuscript.

**Funding:** A.J.L.M and E.C.d.M. were partially supported by IMET and IEMEST. This is IMET contribution number IMET 21-002.

**Institutional Review Board Statement:** Not applicable.

**Informed Consent Statement:** Not applicable.

**Data Availability Statement:** Not applicable.

**Conflicts of Interest:** The authors declare no conflict of interest.

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
