# Peer review of "The Neurochaperonopathies: Anomalies of the Chaperone System with Pathogenic Effects in Neurodegenerative and Neuromuscular Disorders"

_applsci, doi:10.3390/app11030898_

Round 1

Reviewer 1 Report

Very interesting and well written review of the recent literature of chaperone linked diseases. There is a strong connection between the clinical and genetic data that makes this very informative. 

I disagree with the use of the term acquired neurochaperonopathy. The explanation makes sense, but I don't see any data to support the acquired nature of these chaperone changes. I think the authors make excellent points and these sources fit within the scope of the review, but they should consider calling them epigenetic neurochaperonopathy. The literature discussed in this portion focuses on PTMs and epigenetic is a better descriptor of these changes, rather than acquired. 

Minor points:

Consistent use of alpha/α - I noticed the word is still in lines 121, 165. Possibly others. 

Lines 230/258 might needs some formatting so Chaperonin group I/II stands out - looks like a sentence right now. 

Author Response

To: Editors for the Special Issue entitled " Extracellular Chaperones and Related miRNA as Diagnostic Tools of Chronic Diseases" - Applied Sciences.

Subject: Manuscript ID applsci-1073768 titled “The neurochaperonopathies: Anomalies of the chaperone system with pathogenic effects in neurodegenerative and neuromuscular disorders

Dear Editor,

            Thank you very much for your interest in our manuscript. We also thank the Reviewers for their very useful comments and suggestions, which we have followed in preparing the revised manuscript now submitted. We added explanations and changed various parts of the text, as indicated by the Reviewers. All new or modified parts are visible in Track Changes. Below this message, you will find a point-by-point response to the Reviewers’ comments.

We hope the manuscript is now acceptable for publication in Applied Sciences.

Sincerely,

Federica Scalia

Response to Reviewer 1

Reviewer 1 comment #1: Very interesting and well written review of the recent literature of chaperone linked diseases. There is a strong connection between the clinical and genetic data that makes this very informative.

I disagree with the use of the term acquired neurochaperonopathy. The explanation makes sense, but I don't see any data to support the acquired nature of these chaperone changes. I think the authors make excellent points and these sources fit within the scope of the review, but they should consider calling them epigenetic neurochaperonopathy. The literature discussed in this portion focuses on PTMs and epigenetic is a better descriptor of these changes, rather than acquired.

Authors’ Reply: First of all, we thank this Reviewer for the very useful comments, which we have carefully considered in writing the revision of our manuscript. Specially, we appreciate the comment on acquired vs epigenetic terms. Over the last couple of years, we have been discussing about the use of the term acquired as applied to chaperonopathies that are not linked to a variant of a chaperone gene and are not inherited in a Mendelian fashion. One reason was that even those chaperonopathies that are not directly linked to a mutation of a chaperone gene, may be favored by variants of another unrelated gene (or genes) that predispose the patient to the chaperonopathy, and this predisposition could be hereditary. However, in the absence of studies in this area, we decided to continue using the term acquired as described because it is familiar to physicians and pathologists and is used in all medical specialties routinely, in practice and research, not only for chaperonopathies but also for many other pathological conditions.

PTMs are not generally considered epigenetic changes although some epigenetic changes can be caused by PTM of Histones. We believe that at this time it would be misleading to physicians and pathologists, i.e., the intended audience of this article, to call the acquired chaperonopathies caused by PTM of chaperone proteins as epigenetic chaperonopathies. There is no evidence that epigenetic changes regulating the expression of chaperone genes cause the PTMs observed in the chaperone proteins. We suggest that, for the time being, we continue with the use of acquired because it is the accepted term in Medicine to distinguish diseases that are not linked to a genetic, hereditary defect, and are not familial. Of course, this terminology may be changed in the future when more information becomes available about the entire genome of each patient with a chaperonopathy. We thank again this Reviewer for this insightful comment.

Reviewer 1 comment #2:  Minor points:

Consistent use of alpha/α - I noticed the word is still in lines 121, 165. Possibly others.

Authors’ Reply: We wrote α throughout

Lines 230/258 might needs some formatting so Chaperonin group I/II stands out - looks like a sentence right now.

Authors’ Reply: We wrote the subheadings in bold face.

Reviewer 2 Report

In this review, the authors are detailing the various chaperonopathies leading to neurodegenerative and neuromuscular disorders that can arise from abnormal chaperon functions and discuss the possible mechanisms explaining the complicated phenotype-genotype relationships.

Despite the hard work of summarizing current knowledge about the different chaperone mutations and their consequences, it is disconcerting that the aim of this manuscript is not made clear until the very end, and that its structure is not cohesive. The logical flow of the manuscript is currently based on the simple classification of chaperones, although there seems to be little to no connection between sections, which undermines the quality of the manuscript. Therefore I do not think that the current version of the manuscript is meeting the standards to be published in Applied Sciences yet and recommend that the following points are being addressed first.

Major concerns:

  • The authors are not making good use of the limited space of an abstract: most importantly, the aim of the review has to be clarified for readers, and the abstract has to be structured accordingly, which is not the case currently. Please provide the relevant context, the state of current knowledge, what will be the aim and focus of this review and what will be discussed in the end. The abstract should properly end with a strong conclusive statement reflecting the aim of the review. The novelty angle, as well as the relevance of this review, have to be touched upon too. Finally, one cannot indulge to be repetitive in an abstract (as in lines 15-17 and lines 21-23). Too many details dilute the important message. Please adjust accordingly.
  • Similar to the abstract, sections 1 to 4, and especially sections 1 and 2, have to be re-structured with a proper introduction sentence that gives context and relevance in terms of the logical flow running through the manuscript, and that clarifies what will be discussed in that specific paragraph. Together with no clear aims, that makes the manuscript really hard to read and follow through. Additionally, without a proper ending to each paragraph, it really disrupts the logical flow, unless the goal is to simply list the different phenotypes and causes (in that case it would have to be acknowledged at the beginning too). Please re-write accordingly.
    • For example, the idea of the sentence starting on line 108 should actually be at the beginning of that part to explain its purpose.
    • Another example is the idea in the paragraph starting on line 253 that should be placed at the beginning of that part to justify why it is relevant to describe HSPD1/E1.
  • The line between statement and speculation is blurry on many instances (especially in the beginning), such as in the paragraph starting on line 95. At least one research paper should be cited after a statement of fact/result as a reference, and then a review paper can be cited along with the discussion of the result. Citing just the review paper (and by the same authors which makes it worse) suggests it is a speculation while the writing style at the beginning suggests it is a statement. Please add references to research papers for statement finishing line 98 and review the manuscript carefully to avoid this type of inaccuracy.

Minor concerns:

  • Cohesion between different sections could be improved by editing the writing style.
  • Line 258/9: clarify that CCT is a multi-subunit complex that folds protein (lines 282/3), which should not be confused with the CCT family definition
  • Paragraph starting line 392: clarify in which organism those studies were undertaken
  • Lines 412-418: please clarify the link between VCP and histones H3 and H4 deacetylation.
  • Typos:
    • Lines 12/13: the semi-colon is not used correctly
    • Line 16: using post-transcriptional versus post-translational in line 22 is confusing, same in line 40
    • Line 24: Remove "(positive or negative)"
    • Lines 40-42: “its product may (?) undergo a post-translational modification (PTM), which will render the modified protein pathogenic. In either case, aggregates formed by some of the abnormal proteins are cytotoxic and cause disease.”
    • Line 42: Humans
    • Lines 265/268: “Left” appears twice
    • Line 387: oxidative stress induced...
    • Line 394: Clarify “PTMs n”
    • Line 395: please clarify the phosphorylation by a phosphatase?
    • Lines 432/532: pertinent => relevant
    • Line 451: have => has
    • Revise style/grammar/English: lines 48-49, 60-61,65, 247, 397

Author Response

To: Editors for the Special Issue entitled " Extracellular Chaperones and Related miRNA as Diagnostic Tools of Chronic Diseases" - Applied Sciences.

Subject: Manuscript ID applsci-1073768 titled “The neurochaperonopathies: Anomalies of the chaperone system with pathogenic effects in neurodegenerative and neuromuscular disorders

Dear Editor,

            Thank you very much for your interest in our manuscript. We also thank the Reviewers for their very useful comments and suggestions, which we have followed in preparing the revised manuscript now submitted. We added explanations and changed various parts of the text, as indicated by the Reviewers. All new or modified parts are visible in Track Changes. Below this message, you will find a point-by-point response to the Reviewers’ comments.

We hope the manuscript is now acceptable for publication in Applied Sciences.

Sincerely,

Federica Scalia

Response to Reviewer 2

Reviewer 2 comment #1:  In this review, the authors are detailing the various chaperonopathies leading to neurodegenerative and neuromuscular disorders that can arise from abnormal chaperon functions and discuss the possible mechanisms explaining the complicated phenotype-genotype relationships.

Despite the hard work of summarizing current knowledge about the different chaperone mutations and their consequences, it is disconcerting that the aim of this manuscript is not made clear until the very end, and that its structure is not cohesive. The logical flow of the manuscript is currently based on the simple classification of chaperones, although there seems to be little to no connection between sections, which undermines the quality of the manuscript. Therefore I do not think that the current version of the manuscript is meeting the standards to be published in Applied Sciences yet and recommend that the following points are being addressed first.

Major concerns:

The authors are not making good use of the limited space of an abstract: most importantly, the aim of the review has to be clarified for readers, and the abstract has to be structured accordingly, which is not the case currently. Please provide the relevant context, the state of current knowledge, what will be the aim and focus of this review and what will be discussed in the end. The abstract should properly end with a strong conclusive statement reflecting the aim of the review. The novelty angle, as well as the relevance of this review, have to be touched upon too. Finally, one cannot indulge to be repetitive in an abstract (as in lines 15-17 and lines 21-23). Too many details dilute the important message. Please adjust accordingly.

Authors’ Reply: Firstly, we would like to thank this Reviewer for the comments and suggestions, which we have followed in preparing the revision of our manuscript.

We composed a new Abstract following the Reviewer’s suggestion. The Abstracts in Applied Sciences are not structured. Therefore, we composed the new Abstract, including introduction (chaperone system, chaperonopathies, diseases of the central and peripheral nervous systems), objectives, scope, contents (genetic and acquired neurochaperonopathies, genetic variants and post-translation modifications, list of chaperone genes discussed), conclusions and perspectives for the future (improved diagnosis and patient management, and development of chaperonotherapy) in a comprehensive text (208 words) without demarcating the subdivisions with headings.

Reviewer 2 comment #2 Similar to the abstract, sections 1 to 4, and especially sections 1 and 2, have to be re-structured with a proper introduction sentence that gives context and relevance in terms of the logical flow running through the manuscript, and that clarifies what will be discussed in that specific paragraph. Together with no clear aims, that makes the manuscript really hard to read and follow through. Additionally, without a proper ending to each paragraph, it really disrupts the logical flow, unless the goal is to simply list the different phenotypes and causes (in that case it would have to be acknowledged at the beginning too). Please re-write accordingly.

For example, the idea of the sentence starting on line 108 should actually be at the beginning of that part to explain its purpose.

Authors’ Reply: We have added a new introductory paragraph to Section 1. The conclusion paragraph of this Section, which serves as an introduction to what follows, was already in the original manuscript, and we left it untouched because it complies with the Reviewer’s suggestion.

The same applies to Section 2. We inserted a new introductory paragraph at the beginning of Section 2, and the conclusive paragraph was left as in the original version of the manuscript because it complies with the Reviewer’s suggestion.

The other Sections have an introductory portion such as the one suggested by this Reviewer.

Reviewer 2 comment #3 Another example is the idea in the paragraph starting on line 253 that should be placed at the beginning of that part to justify why it is relevant to describe HSPD1/E1.

Authors’ Reply: We have added a phrase at the beginning as required by this Reviewer.

Reviewer 2 comment #4 The line between statement and speculation is blurry on many instances (especially in the beginning), such as in the paragraph starting on line 95. At least one research paper should be cited after a statement of fact/result as a reference, and then a review paper can be cited along with the discussion of the result. Citing just the review paper (and by the same authors which makes it worse) suggests it is a speculation while the writing style at the beginning suggests it is a statement. Please add references to research papers for statement finishing line 98 and review the manuscript carefully to avoid this type of inaccuracy.

Authors’ Reply: The relevant papers are cited in the Table and complete bibliographies are provided in the reviews cited, including the updated website. However, we agree with the Reviewer in that statements should be backed up by pertinent references to original work. We have modified the text to avoid writing a review within a review, and also because we are preparing a review on these issues already committed for a clinical journal.

Reviewer 2 comment #5 Minor concerns:

Cohesion between different sections could be improved by editing the writing style.

Authors’ Reply: done as mentioned above (questions 2 and 3).

Reviewer 2 comment #6 Line 258/9: clarify that CCT is a multi-subunit complex that folds protein (lines 282/3), which should not be confused with the CCT family definition

Authors’ Reply: A new paragraph was added describing the CCT complex.

Reviewer 2 comment #7 Paragraph starting line 392: clarify in which organism those studies were undertaken

Authors’ Reply:. We modified the text clarifying the model used in the work we cited.

Reviewer 2 comment #8 Lines 412-418: please clarify the link between VCP and histones H3 and H4 deacetylation.

Authors’ Reply: We modified the text clarifying the link between VCP and histone H3 and H4 deacetylation.

Reviewer 2 comment #9 Typos:

Lines 12/13: the semi-colon is not used correctly.

Reply: One of the authors specializes in English and we follow the rule that separation of various items including subitems is done with semicolons, while the subitems are separated with commas.

Line 16: using post-transcriptional versus post-translational in line 22 is confusing, same in line 40

Reply: They mean two different things.

Line 24: Remove "(positive or negative)"

Reply: Done as requested.

Lines 40-42: “its product may (?) undergo a post-translational modification (PTM), which will render the modified protein pathogenic. In either case, aggregates formed by some of the abnormal proteins are cytotoxic and cause disease.”

Reply: Done as requested.

Line 42: Humans

Reply: Corrected as indicated.

Lines 265/268: “Left” appears twice.

Reply: Corrected.

Line 387: oxidative stress induced...

Reply: Corrected.

Line 394: Clarify “PTMs n”

Reply: Done.

Line 395: please clarify the phosphorylation by a phosphatase?

Reply: Clarified as requested.

Lines 432/532: pertinent => relevant

Reply: Done.

Line 451: have => has

Reply: Corrected.

Revise style/grammar/English: lines 48-49, 60-61,65, 247, 397

Reply: Done.

Reviewer 3 Report

This is a well-focused, concise, but comprehensive and thorough review of chaperonopathies in neurodegenerative and neuromuscular disorders. It is well structured and written and I have no major concerns about it.

I note two very minor points:

p1l35 "A common feature in neuropathology is protein misfolding with subsequent formation of protein aggregates and precipitates [1–4]. Thus, many neurological disorders can be considered proteinopathies."

The term 'proteinopathy' indicates a condition in which the protein misfolding is causal. As the authors are certainly aware, given the presence of misfolded protein it does not necessarily follow that a disease is caused by protein misfolding (although this certainly the case in some diseases, and may be in others). Therefore, the word 'thus' could be misleading here. The wording should be softened e.g. "A common feature in neuropathology is protein misfolding with subsequent formation of protein aggregates and precipitates [1–4], and a number of neurological disorders may be proteinopathies or have a proteinopathy component."

p6l138: muscles weakness > muscle weakness (please check for similar typos throughout)

Author Response

To: Editors for the Special Issue entitled " Extracellular Chaperones and Related miRNA as Diagnostic Tools of Chronic Diseases" - Applied Sciences.

Subject: Manuscript ID applsci-1073768 titled “The neurochaperonopathies: Anomalies of the chaperone system with pathogenic effects in neurodegenerative and neuromuscular disorders

Dear Editor,

            Thank you very much for your interest in our manuscript. We also thank the Reviewers for their very useful comments and suggestions, which we have followed in preparing the revised manuscript now submitted. We added explanations and changed various parts of the text, as indicated by the Reviewers. All new or modified parts are visible in Track Changes. Below this message, you will find a point-by-point response to the Reviewers’ comments.

We hope the manuscript is now acceptable for publication in Applied Sciences.

Sincerely,

Federica Scalia

Response to Reviewer 3

Reviewer 3 comment #1: This is a well-focused, concise, but comprehensive and thorough review of chaperonopathies in neurodegenerative and neuromuscular disorders. It is well structured and written and I have no major concerns about it.

I note two very minor points:

p1l35 "A common feature in neuropathology is protein misfolding with subsequent formation of protein aggregates and precipitates [1–4]. Thus, many neurological disorders can be considered proteinopathies."

The term 'proteinopathy' indicates a condition in which the protein misfolding is causal. As the authors are certainly aware, given the presence of misfolded protein it does not necessarily follow that a disease is caused by protein misfolding (although this certainly the case in some diseases, and may be in others). Therefore, the word 'thus' could be misleading here. The wording should be softened e.g. "A common feature in neuropathology is protein misfolding with subsequent formation of protein aggregates and precipitates [1–4], and a number of neurological disorders may be proteinopathies or have a proteinopathy component."

Authors’ Reply: We thank the Reviewer for the comments, which helped us to improve the manuscript. We modified the text according to his suggestion.

Reviewer 3 comment #2:  p6l138: muscles weakness > muscle weakness (please check for similar typos throughout)

Authors’ Reply: We corrected the typos (although as the Reviewer must know, they are little demons that easily escape the authors but appear in force to other readers…We hope that in the end with the help of Reviewers, Editors, and typesetters, their number will be reduced to zero).
